# D-Loop Mutation G42A/G46A Decreases Actin Dynamics

**DOI:** 10.3390/biom10050736

**Published:** 2020-05-08

**Authors:** Mizuki Matsuzaki, Ikuko Fujiwara, Sae Kashima, Tomoharu Matsumoto, Toshiro Oda, Masahito Hayashi, Kayo Maeda, Kingo Takiguchi, Yuichiro Maéda, Akihiro Narita

**Affiliations:** 1Graduate School of Science, Nagoya University, Furo-cho, Chikusa-ku, Nagoya 464-8601, Japan; matsuzaki.mizuki.nagoya@gmail.com (M.M.); skashima305@yahoo.co.jp (S.K.); t_matumoto@nagoya-u.jp (T.M.); maeda.kayo@h.mbox.nagoya-u.ac.jp (K.M.); j46037a@cc.nagoya-u.ac.jp (K.T.); 2Graduate School of Science, Osaka City University, Sugimoto, Sumiyoshi-ku, Osaka 558-8585, Japan; ikuko.fujiwara@sci.osaka-cu.ac.jp; 3Faculty of Health and Welfare, Tokai Gakuin University, 5-68 Nakakirino-cho, Kakamigahara, Gifu 504-8511, Japan; toda375@tokaigakuin-u.ac.jp; 4Department of Frontier Bioscience, Hosei University, 3-7-2 Koganei-cho, Koganei, Tokyo 184-8584, Japan; hayashi.bmi@gmail.com; 5Graduate School of Informatics, Nagoya University, Furo-cho, Chikusa-ku, Nagoya 464-8601, Japan; ymaeda@cc.nagoya-u.ac.jp

**Keywords:** depolymerization, polymerization, intrinsically disordered, cofilin

## Abstract

Depolymerization and polymerization of the actin filament are indispensable in eukaryotes. The DNase I binding loop (D-loop), which forms part of the interface between the subunits in the actin filament, is an intrinsically disordered loop with a large degree of conformational freedom. Introduction of the double mutation G42A/G46A to the D-loop of the beta cytoskeletal mammalian actin restricted D-loop conformational freedom, whereas changes to the critical concentration were not large, and no major structural changes were observed. Polymerization and depolymerization rates at both ends of the filament were reduced, and cofilin binding was inhibited by the double mutation. These results indicate that the two glycines at the tip of the D-loop are important for actin dynamics, most likely by contributing to the large degree of conformational freedom.

## 1. Introduction

Actin is an abundant protein in eukaryotes that forms a double-stranded filament. α-Skeletal and β-cytoplasmic actins from mammals share 100% identical sequences with birds and reptiles, indicating the importance of actin in the cell. Actin plays central roles in a wide range of cellular functions, including cell motility, the cytoskeleton, cell division and muscle contraction [1]. In most cases, actin dynamics through depolymerization and polymerization are indispensable.

The monomeric state (G-form) and fibril state (F-form) of actin differ substantially in structure [2]. Actin contains two large rigid bodies [2,3]. There are two prominent structural differences between the G-form and F-form of actin. One is the relative orientation between the two rigid bodies and the other is the conformational change to the DNase I binding loop (D-loop) [4] (Figure 1e).

The D-loop is an intrinsically disordered loop with a large degree of conformational freedom. In many crystals, the density from the D-loop cannot be observed. Only in limited crystals (~50 out of 250 crystals analyzed in [2]) or in actin filaments [6,12,13], when the D-loop binds to another protein or subunit, has the D-loop been observed. Even among crystal structures with observable D-loops, the structure of this loop adopts a range of conformations (Figure 1g), indicating that it samples a large conformational space with a high degree of freedom. These observations indicate that the D-loop cannot form a certain conformation in the absence of a binding partner, and therefore the D-loop can be considered as an intrinsically disordered protein region. Interestingly, the sequence of the D-loop is well conserved among species (Figure 1b). This sequence conservation indicates that the extent of conformational freedom of the D-loop is also conserved, which may contribute to intrinsic actin dynamics. There are three glycines in the D-loop that are present in all known actin sequences (Figure 1b) and likely contribute largely to the conformational freedom of the D-loop because glycine can sample the largest conformational space among the 20 amino acids found in proteins [14,15].

Cleavage [16] and oxidation [17] of the D-loop reduces the stability of the actin filament significantly because the D-loop forms part of the subunit–subunit interface [12,18]. Spin-label and fluorescent experiments have shown that the D-loop adopts multiple conformations even in the actin filament [19]. However, the role that D-loop conformational freedom plays in actin dynamics remains unclear. Here, we mutated two glycines in the D-loop to alanine (G42A/G46A). Introduction of the two alanine side chains prohibited some conformations that are possible for the wild-type protein and restricted the conformational freedom of the D-loop. We measured the effects of the mutations on the critical concentration, actin filament structure, actin dynamics and cofilin binding. The results indicated that the two glycines are important in determining the turnover rates of actin dynamics in the cell, most likely by increasing the extent of conformational freedom of the D-loop, although additional mutant experiments are required to confirm this hypothesis.

## 2. Materials and Methods

### 2.1. Construction of the β-Actin Expression Vector

The human β-actin constructs contained a Strep-Tag II sequence (WSHPQFEK), spacer sequence (SS) and the tobacco etch virus (TEV) protease recognition sequence (GRENLFQ) at the N-terminus of the constructs. The corresponding amino acid sequence at the N-terminus is “MGWSHPQFEKSSGRENLYFQMDDDIAAL…” (the non-native protein sequence is underlined). This human β-actin gene was cloned into the pFastBac vector (Invitrogen, Thermo Fisher Scientific, Waltham, MA, USA) at BamHI and EcoRl restriction sites to give the pFastBac-L21-WT expression plasmid. L21 is an enhancer sequence [20]. The double mutant actin, G42A/G46A, was generated using the KOD Plus Mutagenesis Kit (Toyobo, Osaka, Japan) and the pFastBac-L21-WT expression plasmid as the template. The Bac-to-Bac system (Invitrogen, Thermo Fisher Scientific, Waltham, MA, USA) was used to obtain the recombinant virus. Primers for generating the coding sequence of wild-type β-actin: 5’-CATCGGGCGCGGATCCATGGGATGGTCGCATCAGTTC-3’ and 5’-GTAGGCCTTTGAATTCTAGAAACACTTCCTGACG-3’; primers for generating the coding sequence of the G42A/G46A mutant actin: 5’-GCAGTGATGGTCGTCATGGGCCAAAAG-3’ and 5’-CTGGTGGCGTGGACGGCCAAC-3’.

### 2.2. Expression of Recombinant Human β-Actin in Insect Cells

Sf9 insect cells were used to express human β-actin. Sf9 cells were cultured in Grace’s insect medium supplemented (Invitrogen, Thermo Fisher Scientific, Waltham, MA, USA) with 10% fetal bovine serum, 50 U mL^−1^ penicillin G, 50 µg mL^−1^ streptomycin and 0.2% Pluronic F-68 at 27 °C with a total volume of 200 mL. After 72 h post-transfection, the cells were harvested by centrifugation and stored at −20 °C until use.

### 2.3. Purification of Recombinant Actins

The infected cells were lysed in extraction buffer (20 mM Tris-HCl, 0.2 mM CaCl_2_, 0.2 mM adenosine triphosphate (ATP), 1 mM dithiothreitol (DTT), complete EDTA-free Protease Inhibitor Cocktail (Roche Applied Science, Penzberg, Germany), pH 8.0). The cell lysate was sonicated and then centrifuged at 32,000× *g* for 30 min. The supernatant was loaded onto a StrepTrap HP column (GE Healthcare, Chicago, IL, USA), washed with the extraction buffer (without protease inhibitor) and the target protein eluted with the elution buffer (20 mM Tris-HCl, 0.2 mM CaCl_2_, 0.2 mM ATP, 1 mM DTT and 2.5 mM desthiobiotin, pH 8.0). The eluted fraction was mixed with G-Buffer (2 mM Tris-HCl, 0.2 mM CaCl_2_, 0.2 mM ATP, 0.2 mM DTT, pH 8.0) to prevent polymerization and concentrated using Amicon Ultra-15 Centrifugal Filter Units (30,000 NMWL, Merck KGaA, Darmstadt, Germany). The fraction was polymerized by adding 100 mM KCl and 2 mM MgCl_2_ and then dialyzed against F-buffer (2 mM Tris-HCl, 100 mM KCl, 2 mM MgCl_2_, 0.2 mM ATP, 0.2 mM DTT, pH 8.0) for more than 9 h. The N-terminally tagged F-actin was collected by centrifugation at 451,000× *g* for 30 min at 4 °C. The N-terminally tagged F-actin was resuspended in G-buffer and then dialyzed against G-Buffer at 4 °C for more than 9 h. The dialyzed solution was centrifuged at 451,000× *g* for 30 min. The supernatant was diluted with G-buffer (final concentration of actin was 12 μM), and the Strep-Tag II was cleaved by TurboTEV protease (Accelagen, San Diego, CA, USA). The sample was loaded onto a StrepTrap HP column to remove tag-G-actin. Native PAGE was used to confirm that tag-G-actin was removed (Figure 1d). The flow-through fraction was polymerized by the addition of 100 mM KCl and 2 mM MgCl_2_. F-actin was collected by centrifugation at 451,000× *g* for 30 min at 4 °C. The F-actin pellet was then resuspended in G-buffer and dialyzed against G-buffer at 4 °C for more than 9 h. The dialyzed solution was centrifuged at 451,000× *g* for 30 min at 4 °C and the resulting supernatant fraction was used as purified recombinant actin. The final yield of the protein was ~0.1 mg per 100 mL culture for wild-type actin and ~0.05 mg per 100 mL culture for the G42A/G46A mutant. This small yield of protein restricted possible experiments.

### 2.4. Native-PAGE

The BIO CRAFT BE-210 system (Bio Craft, Tokyo, Japan) was used to perform Native-PAGE. The running gel contained 10% acrylamide/bisacrylamide (a mixture at a ratio of 37.5:1) in 375 mM Tris-HCl (pH 8.8), 0.2 mM ATP, 0.3 mM CaCl_2_ and 1 mM DTT. The stacking gel contained 4.8% acrylamide/bisacrylamide (a mixture at a ratio of 37.5:1) in 125 mM Tris-HCl (pH 6.8), 0.2 mM ATP, 0.3 mM CaCl_2_ and 1 mM DTT. The gels were bathed in running buffer (25 mM Tris, 250 mM glycine, 0.2 mM ATP, 0.3 mM CaCl_2_, 1 mM DTT) and samples (20 pmol per lane mixed with the same volume of 2× loading buffer (4 mM Tris-HCl, 0.4 mM ATP, 0.6 mM CaCl_2_, 2 mM DTT, 10% (*w*/*v*) sucrose, pH 8.0)) were run at 100 V for 30 min and thereafter at 200 V for 30 min at room temperature.

### 2.5. Western Blotting

Recombinant actin samples (Tag-G-actin, TEV added Tag-G-actin, purified G-actin) and skeletal muscle actin purified from chicken [21] as a marker were electrophoresed by native-PAGE and then transferred onto a Hybond-ECL membrane (GE Healthcare, Chicago, IL, USA) by using a Mini-Protean II Cell with a Mini Trans Blot Cell (Bio-Rad, Hercules, CA, USA) at 33 mA for 3 h. The membrane was blocked with 5% skim milk in TBS-T (Tris Buffered Saline with Tween-20) at 4 °C overnight. After incubation with the strep-tag antibody (mouse monoclonal, Qiagen, Hilden, Germany) at room temperature for 1 h, the membrane was incubated with an anti-mouse IgG (γ-chain specific)-peroxidase antibody produced in goat (Sigma-Aldrich, St. Louis, MO, USA) at room temperature for 1 h. Protein bands were detected by the Amersham ECL Western blotting analysis system kit (GE Healthcare, Chicago, IL, USA).

### 2.6. Critical Concentrations

Actin was polymerized in F-buffer for 60 min at room temperature. The final concentrations of the recombinant actin were 0.5–3 μM. Forty microliters of the samples were centrifuged at 451,000× *g* (k-value = 7) for 30 min to harvest polymerized actin. The harvested actin was resuspended in 40 μL F-buffer. The supernatant and the resuspended pellet were mixed with sample buffer (a mixture of NuPAGE LDS (lithium dodecyl sulfate) Sample Buffer (4×) (Thermo Fisher Scientific, Waltham, MA, USA), 1 M DTT and ultra-pure water at a ratio of 15:6:19) and 20 μL of the samples were applied to SDS-PAGE gels. The concentration of actin in the supernatant was measured by densitometry of the actin band in the SDS-PAGE gel. We confirmed that the critical concentration was independent of the actin concentration over the range of 0.5–3 μM.

### 2.7. Electron Microscopy

Actin was polymerized in F-buffer for 60 min at room temperature. The actin filaments fully decorated with cofilin (cofilactin) were polymerized by the same procedures as described in the “Co-sedimentation assay” (see below), except for the final cofilin concentration: 12 μM was used instead of 2 μM. Polymerized samples (each 2.0 μL) were applied onto the grid (#10-1012 ELS-C10, Okenshoji, Tokyo, Japan), tobacco mosaic virus (2.0 µL, 0.03 mg/mL) was added to stain the grid uniformly and the sample was negatively stained with uranyl acetate. Electron micrographs of the actin filament were recorded on electron microscopic film FG (Fujifilm, Tokyo, Japan) at a magnification of 40,000 by using a H-7650 transmission electron microscope (TEM) (Hitachi High-Technologies, Tokyo, Japan) operated at 100 kV. The film was digitized with a GT-X970 scanner (Epson, Suwa, Japan) at a resolution corresponding to 0.26 nm/pixel. Cofilactin grids were imaged by a SU9000 scanning transmission electron microscope (STEM) (Hitachi High-Technologies, Tokyo, Japan) at 0.41 nm/pixel operated at 30 kV.

### 2.8. Proteins and Protein Labeling for Total Internal Reflection Fluorescence (TIRF) Microscopy

Rabbit skeletal actin used for labeling was obtained from an acetone powder of rabbit skeletal muscle that was purified by repeated polymerization and depolymerization and stored in the G_TIRF_ buffer (10 mM HEPES, 0.2 mM ATP, 0.1 mM CaCl_2_, pH 7.5) at 4 °C [21]. The purified skeletal muscle ATP-actin was labeled with Alexa Fluor 488 succinimidyl ester (Thermo Fisher Scientific, Waltham, MA, USA, #20100), as described previously [22]. The labeled fraction of actin was estimated by measuring the absorbance at 290 nm for actin and 491 nm for Alexa Fluor 488. Concentrations of actin and the fluorescence dye were determined by using the molar extinction coefficients of 26,600 M^−1^ cm^−1^ at 290 nm and 96,900 M^−1^ cm^−1^ at 491 nm, respectively. The labeling ratio was estimated to be higher than 98%. The labeled actin was prepared by mixing 20% Alexa Fluor 488-skeletal muscle actin with 80% recombinant β-actin in the G_TIRF_ buffer. The TIRF polymerization assay was performed at room temperature and the polymerization buffer consisted of 50 mM KCl, 1 mM MgCl_2_, 1 mM EGTA (ethylene glycol tetraacetic acid), 10 mM imidazole, 100 mM DTT, 0.2 mM ATP and 0.5% (*w*/*v*) methylcellulose (1500 cP from Sigma-Aldrich, St. Louis, MO, USA). One hundred millimolar DTT has been used with various papers to observe individual actin filaments and to avoid photo bleaching [23,24,25].

### 2.9. TIRFAssay and Image Analysis

Protocols for the TIRF assay have been described previously [26]. In brief, skeletal muscle myosin was inactivated by treating with 50 nM N-ethylmaleimide (NEM) [23] and loaded into an observation chamber to tether actin filaments. Unbound NEM-myosin was removed by washing with 1× KMEI buffer (50 mM KCl, 1 mM EGTA, 1 mM MgCl_2_, 10 mM imidazole-HCl, pH 7.0). The sample was loaded into the observation chamber immediately after initiating actin polymerization by adding the actin monomer to the polymerization buffer. Once the fast growing (barbed) end of individual actin filaments were distinguishable and filaments had elongated to lengths of ~10 µm, free actin monomers in the observation chamber were removed by loading the polymerization buffer to initiate depolymerization. Time course observations were performed by an objective TIRF microscope (Ti-2000, Nikon, Tokyo, Japan) with an EMCCD (iXON3, Andor, Oxford instruments, Abingdon, UK). Each image was captured from an exposure time of 500 ms and samples were kept in the dark except when images were taken at 15 s intervals. ImageJ with a plugin that was developed by Jeffrey Kuhn (Massachusetts Institute of Technology, Cambridge, MA, USA) was used to measure changes in the length of the actin filaments. Average rates of polymerization and depolymerization were estimated from the mean of the histograms of the change in length at intervals of 15 s in each polymerization and depolymerization phase.

### 2.10. Co-Sedimentation Assay

Chicken cofilin was expressed and prepared as described previously [3]. Recombinant actin was polymerized in the cofilactin buffer (15 mM PIPES-KOH, 50 mM KCl, 2 mM MgCl_2_, 2 mM ATP, pH 6.6) for 90 min at room temperature. The polymerized actin was added to cofilin in the cofilactin buffer and incubated on ice for 10 min. The final concentrations of the recombinant actin and cofilin were 2 μM and 2 μM, respectively. Forty microliters of the sample was then centrifuged at 451,000× *g* for 15 min at 4 °C.

Centrifuged samples were separated into supernatant and pellet. The pellet was resuspended in 40 μL of the cofilactin buffer. The supernatant and the resuspended pellet were mixed with sample buffer (a mixture of NuPAGE LDS Sample Buffer (4×) (Thermo Fisher Scientific, Waltham, MA, USA), 1 M DTT and ultra-pure water at a ratio of 15:6:19) and then 20 μL of the samples were load onto an SDS-PAGE gel. The amount of cofilin bound to the actin filament in the pellet was determined by densitometry of the cofilin band on the SDS-PAGE gel and by using the prepared calibration curve. We also loaded cofilin at concentrations ranging between 1 and 18 μM to lanes of the SDS-PAGE gel for the co-sedimentation assay. The calibration curve for determining the concentration of cofilin present in the actin filament samples was based on the density values of these bands.

### 2.11. Image Analysis

All image analyses were performed by a software package, EOS [27]. Actin and cofilactin filament images were extracted from electron micrographs. The filament images were traced and straightened on a computer. The averaged power spectrum was calculated from the straightened filaments. The straightened filaments were filtered by a bandpass filter, which passes regions around typical four layer lines on the diffraction pattern [28].

## 3. Results

### 3.1. Expression and Purification of Recombinant Human Cytoskeletal β-Actins in Insect Cells

Recombinant human cytoskeletal β-actin constructs with an affinity tag, Strep-Tag II sequence (WSHPQFEK), spacer sequence (SS) and TEV protease recognition sequence (GRENLYFQ) at the N-terminus (Figure 1a) were expressed by using a baculovirus-based expression system in insect cells (Figure 1c). The tag was cleaved by TEV protease and removed at the end of the purification procedure (Figure 1d). Wild-type and the G42A/G46A mutant were expressed and purified (Figure 1c). G42 and G46 are two of the three conserved glycines in the D-loop (Figure 1b,f). Construction of a G42A/G46A mutant model based on the current actin subunit–subunit interaction structure (PDB ID: 6KP8 (Takeda et al., submitted)) revealed that the alanine side chains did not clash with other regions of the actin subunits (Figure 1h). Thus, the alanine mutations were hypothesized to decrease conformational freedom of the D-loop without affecting subunit–subunit interactions.

### 3.2. Effects of the G42A/G46A Mutations on Subunit–Subunit Interactions

We inspected two parameters to confirm that the mutations did not affect the subunit–subunit interactions significantly: (i) the free energy difference between the filament (F-actin) and the monomers (G-actin) and (ii) the structure of the actin filament.

The monomer concentration is independent from the total concentration of actin when the actin solution is in equilibrium between F-actin and G-actin [29]. This monomer concentration was termed the critical concentration, reflecting the free energy difference between the F-actin and the G-actin directly. Poor sample yields hampered evaluation of the critical concentration by light scattering analysis [30] or the pyrene fluorescent assay [31]. Instead, we separated G-actin from F-actin by ultracentrifugation and measured the actin concentration in the supernatant, which reflects the critical concentration, by densitometry of the actin band on an SDS-PAGE gel (Figure 2). Although, in this assay, a small amount of G-actin can be pelleted and a fraction of very short actin filaments can remain in the supernatant, this approach is still valid for comparing the two actin species under the same conditions.

The critical concentrations of the recombinant wild-type and mutant actins were measured four times independently. The ratio of the critical concentration of the mutant against wild-type actin was 0.82 ± 0.03 (standard error). The results thus show no large difference; however, the difference was statistically significant (*p* < 0.0001) because of the small error.

The structure of the actin filament was also evaluated by electron microscopy (Figure 3). Fifteen and 18 filament images of the wild-type and mutant, respectively, were extracted (Figure 3c,d) from electron micrographs of negatively stained specimens (Figure 3a,b). The diffraction was then averaged (Figure 3g,h). We observed identical layer line patterns indicating the nominal helical structure of the actin filament [18,32] in the averaged diffraction patterns for the wild-type and mutant actins. We could not observe any differences in the structures, although small structural differences may exist that are difficult to detect by the current experiments.

### 3.3. Effects of the G42A/G46A Mutations on Polymerization and Depolymerization Rates

We investigated how the mutations affect the polymerization and depolymerization dynamics of actin. Because of the low yield of recombinant actin and to avoid recombinant actin loss from the labeling procedure, we prepared ~98% Alexa Fluor 488-labeled rabbit skeletal muscle actin for visualization under the TIRF microscope. We mixed the labeled rabbit skeletal muscle actin and the recombinant actin at a ratio of 1:4. After initiating actin polymerization by adding the polymerization buffer, the sample was loaded into the observation chamber and the elongation process was monitored (Figure 4a–d, Appendix A). Actin filaments containing 20% labeled skeletal muscle actin were not illuminated uniformly (Figure 4a,b). This non-uniform fluorescence intensity along an actin filament suggests that the labeled skeletal muscle actin may be less incorporated into filaments formed from the recombinant actin. Actin depolymerization was initiated by washing away the free actin monomer in the observation chamber with the polymerization buffer at the time indicated by the dashed lines in Figure 4c,d. The time course plot of length change at each end shows that actin polymerizes and depolymerizes rapidly from one (barbed) end but not from the other (pointed) end (Figure 4c,d). We estimated the polymerization and depolymerization velocities from linear fitting of each actin length profile. The average velocities of polymerization and depolymerization at every actin concentration were estimated from the slopes of linear fittings of more than 10 actin filaments under each condition (Figure 4c,d). By plotting polymerization velocities over actin concentrations (Figure 4e,f), the elongation (*k*^+^) and dissociation (*k*^−^) rates were estimated for the mutant and wild-type actins (Table 1). The elongation rate from the barbed-end of the mutant actin (7.1 µM^−1^ s^−1^) was slower when compared with that of the wild-type recombinant actin (11.0 μM^−1^ s^−1^), which had a near identical rate as skeletal muscle actin from the barbed-end (11 μM^−1^ s^−1^ [33]). This result indicates that D-loop mutations contribute to rapid actin assembly. By removing (through washing) free actin monomers, spontaneous dissociation from each actin filament was initiated and monitored over time. The dissociation rate from the barbed-end of recombinant actin (1.8 s^−1^) was three-fold slower than skeletal muscle actin (5.3 s^−1^ [34]) and approximately three-fold faster when compared with that of the D-loop mutant (0.6 s^−1^), showing that the D-loop also plays a functional role in actin dissociation. These slow dissociation rates at the barbed-end yielded a smaller critical concentration (Cc) for the mutant actin (0.09 μM) when compared with that of the wild-type protein (0.17 μM). The ratio of the two critical concentrations (mutant/WT) was 0.5 ± 0.2, which is not statistically different from the ratio of critical concentrations determined by the sedimentation assay (~0.8, Figure 2) considering the relatively large error (*p* > 0.1, assuming the ratio follows a normal distribution). The elongation rate and the dissociation rate of the mutant at the pointed-end were also slower than wild-type actin; however, the difference in the dissociation rate is not statistically significant because of the difficulty in measuring the slow dynamics at the pointed-end. In conclusion, the results of the TIRF assay showed that polymerization and depolymerization of actin becomes substantially slower when the glycines in the D-loop are mutated.

### 3.4. Effects of the G42A/G46A Mutations on Cofilin Binding

Cofilin binding was also affected by the mutation. Cofilin binding was measured by the sedimentation assay. An actin and cofilin mixture was ultracentrifuged and cofilin bound to the actin filament was found in the pellet. The amount of bound cofilin was measured by the densitometry of the cofilin band in an SDS-PAGE gel (Figure 5a). Cofilin binding was reduced significantly by the G42A/G46A mutations (Figure 5b). The actin filament structure fully decorated with cofilin (cofilactin) was also investigated by electron microscopy. The averaged layer line pattern of the mutant cofilactin was identical to that of the wild-type cofilactin, showing a tighter helix than the actin filament [3,35]. This observation shows that the structure of cofilactin was not affected by the mutation at the current resolution (Figure 6); however, the D-loop mutations reduced cofilin binding.

## 4. Discussion

We mutated two glycines at the tip of the D-loop of actin. The mutations caused only a small change in the critical concentration (reduced by 20%; Figure 2) and differences in the actin filament structure were not observed (Figure 3). The mutations significantly reduced the rates of depolymerization and polymerization of actin at the barbed end and the polymerization at the pointed end when compared with that of wild-type actin, but the difference of the depolymerization rate at the pointed end was not statistically significant between the mutant and wild-type because of the slow dynamics. Considering that the mutations do not seem to affect the subunit–subunit interface (Figure 1h), the different rates are most likely cause by restricted conformational freedom of the D-loop induced by the mutations, although additional mutant experiments are required to confirm this because only one mutant could be investigated in this study due to the difficulties in purifying satisfactory amounts of mammalian actin mutants.

Cofilin binding was also reduced by the mutation, which is consistent with a recently published cofilin-binding model [3,36], where it was proposed that cofilin binding requires spontaneous dissociation of the D-loop from the adjacent subunit. Therefore, the association rate of cofilin is most likely reduced by the D-loop mutations because of restricted conformational freedom. In contrast, the dissociation rate of cofilin from the actin filament should not be affected by the mutations because the D-loop has no contacts with other parts of the filament in cofilactin [4]. Overall, cofilin binding was reduced thermodynamically by the mutations. Cofilin is crucial for accelerating actin dynamics in the cell by disassembling actin filaments [37]. Therefore, the current results indicate that the two glycines at the tip of the D-loop control the speed of actin dynamics in the cell directly, by controlling polymerization, depolymerization rates and cofilin binding.

Actin dynamics have been discussed in the context of ATPase [38,39,40] and the contribution of the D-loop in actin dynamics has remained unresolved. The current results suggest that the large conformational space sampled by the D-loop apart from ATPase contributes to the speed of actin dynamics. The D-loop is an intrinsically disordered loop. There is accumulating evidence indicating the importance of intrinsically disordered proteins/protein regions [41,42]. However, there is a paucity of direct evidence showing that the extent of conformational freedom is important for protein function. Our current results might be a good example of how conformational freedom of an intrinsically disordered protein region affects protein function.

## 5. Conclusions

This study indicates that the two glycines at the tip of the D-loop regulate the speed of actin dynamics in the cell directly by controlling polymerization/depolymerization rates and cofilin binding. Because the mutations do not seem to affect the subunit–subunit interface, it is most likely that the extent of conformational freedom of the D-loop affects the rates and cofilin binding. Actin dynamics have been considered in the context of ATPase. However, our results clearly revealed the importance of the D-loop for actin dynamics.

## Figures and Tables

**Figure 1 biomolecules-10-00736-f001:**
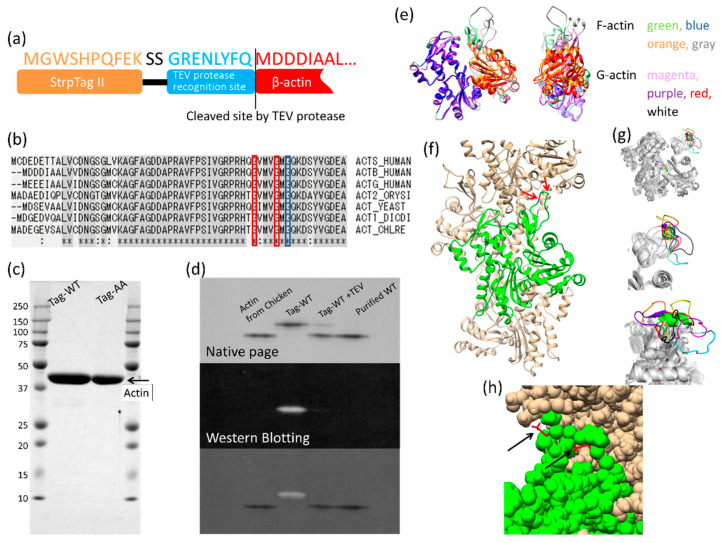
The G42A/G46A double mutation. (**a**) A schematic of recombinant actin with the N-terminal tag. (**b**) Sequence alignment of the N-terminal region of actin. ACTS_HUMAN, ACTB_HUMAN, ACTG_HUMAN, ACT2_ORYSI, ACT_YEAST, ACT1_DICDI and ACT_CHLRE represent α-skeletal muscle actin from human, cytoplasmic β-actin from human, cytoplasmic γ-actin from human, actin-2 from *Oryza sativa* subsp. Indica, actin 1 from yeast, actin from *Dictyostelium discoideum* and actin from *Chlamydomonas reinhardtii*, respectively. Red boxes indicate G42 and G46 of the cytoplasmic β-actin and the corresponding conserved glycines of other actins. The blue box indicates G48 of the cytoplasmic β-actin and the corresponding conserved glycine of the other actins. (**c**) An SDS-PAGE gel of purified recombinant actin with the N-terminal tag. Wild-type (WT) and G42A/G46A mutant (AA) are shown in the second and the third lanes from the left, respectively. Lanes 1 and 4 are molecular mass standards with molecular weights in kilo-Daltons shown on the left-hand side. (**d**) Cleavage of the tag. The upper panel shows a native-PAGE gel. The native-PAGE gel is required to discriminate actin without the tag from that with the tag. Actin from chicken, purified recombinant wild-type (WT) actin with the N-terminal tag (Tag-WT), purified recombinant WT actin with the tag processed by TEV (Tag-WT + TEV; TEV: tobacco etch virus) and the final product (Purified WT) are shown in the lanes from the left. The middle panel shows the result of a Western blot using an antibody for the tag. The lower panel shows a superposition of the images in the upper and middle panels. (**e**) Comparison between G-actin [5] (PDB ID: 1J6Z) and F-actin [6] (PDB ID: 6DJN). Actin contains two rigid bodies: the inner domain (ID) and outer domain (OD) [3]. ID and OD rigid bodies, the D-loop (41–50) and the other parts of F-actin are shown in blue, orange, gray and red, respectively. ID and OD rigid bodies, D-loop and the other parts of G-actin were colored in purple, red, white and magenta, respectively. The ID rigid bodies of G- and F-actin were aligned. The relative orientation of the OD rigid body differs between G- and F-actin without changing the rigid body structure. The structure of the D-loop also clearly differs between G- and F-actin. (**f**) Actin subunit–subunit interactions in one strand (6KP8, Takeda et al., submitted). One subunit and the other subunits are presented in green and camel, respectively. Red arrows indicate locations in the structure of G42 and G46. (**g**) Conformational space of the D-loop. Seven crystal structures with the D-loop were superpositioned by OD rigid body [2,3], and the D-loop (41–50) [4] is presented in different colors: PDB ID: 1J6Z [5] (green), 6FM2 [7] (cyan), 1IJJ [8] (magenta), 4JHD [9] (yellow), 4B1Z [10] (black), 1ATN [4] (purple), 6KP8 (orange) and 3DAW [11] (red). Upper panel: whole structure of actin. Middle panel: an enlarged view of the D-loop. Lower panel: a rotated view by 90° of the middle panel. (**h**) An enlarged view of the actin subunit interactions around the D-loop with side chains represented in the space-filling model. G42 and G46 of 6KP8 were simply replaced by alanines and are shown in red stick models (black arrows). The mutated alanine side chains do not sterically clash with other regions of the actin subunits.

**Figure 2 biomolecules-10-00736-f002:**
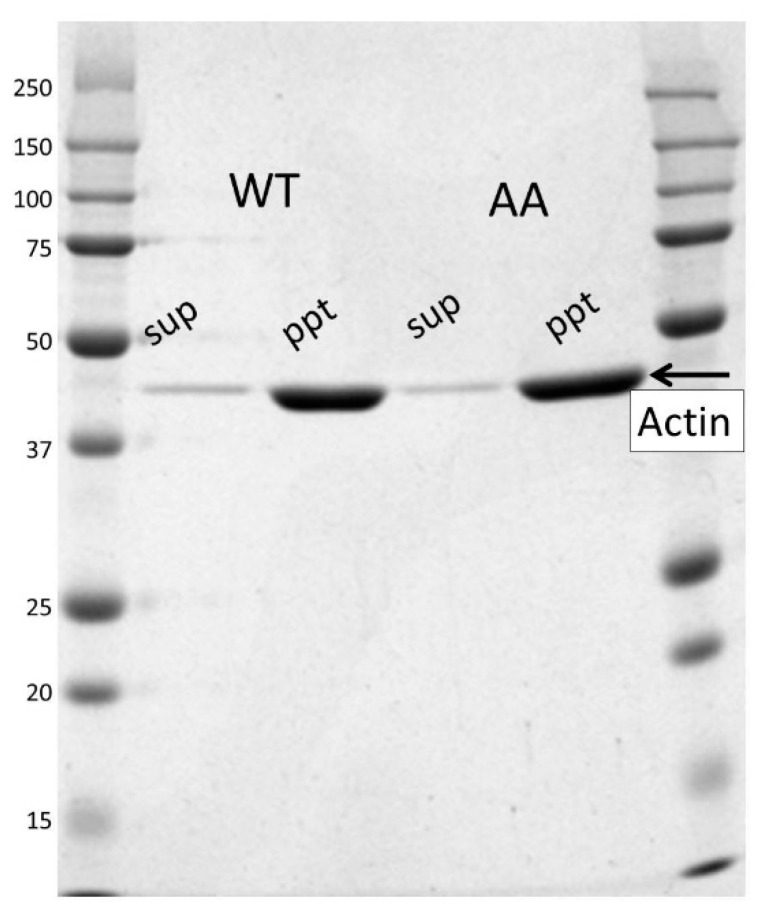
Critical concentration measurement of the actin filament. Recombinant actin (0.5–3 µM as final concentrations) was polymerized in F-buffer (2 mM Tris-HCl, 100 mM KCl, 2 mM MgCl_2_, 0.2 mM ATP (adenosine triphosphate), 0.2 mM DTT (dithiothreitol), pH 8.0) by incubating for 60 min at room temperature and separating by ultracentrifugation into the pellet (ppt) and supernatant (sup). An example of an SDS-PAGE gel for wild-type (WT) and the G42A/G46A mutant (AA) is shown. The critical concentration was measured by densitometry of actin in the supernatant. Four independent experiments were performed and the ratio of the critical concentration of the mutant against the WT was 0.82 ± 0.03 (standard error). The left and right lanes are molecular mass markers with molecular weights provided in kilo-Daltons on the left-hand side.

**Figure 3 biomolecules-10-00736-f003:**
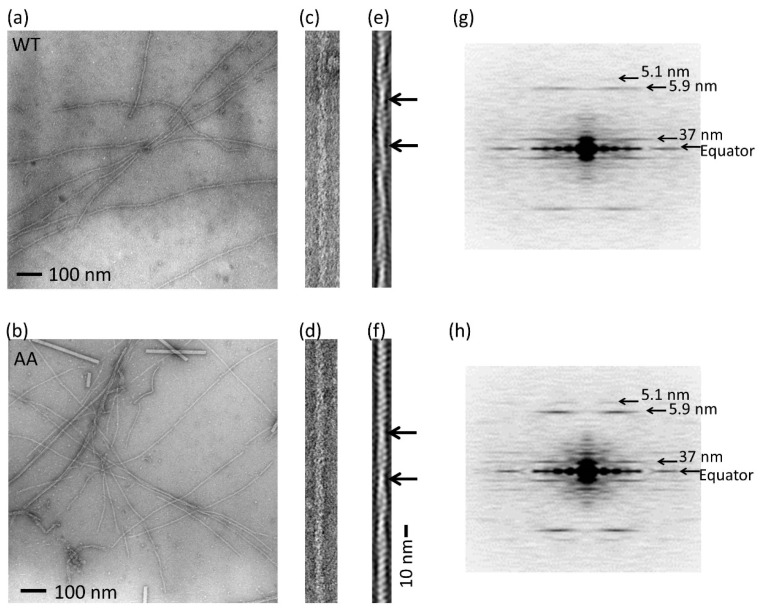
Actin filament structure. (**a**) An example of an electron micrograph of the wild-type (WT) actin filament. (**b**) An example of an electron micrograph of the G42A/G46A mutant actin filament (AA). Scale bars for (**a**) and (**b**) represent 100 nm. (**c**) An enlarged view of the WT actin filament. (**d**) An enlarged view of G42A/G46A mutant actin filament. (**e**) Filtered image of (**c**) by using a bandpass filter, which passes regions around typical four-layer lines on the diffraction pattern indicated by arrows in (**g**) and (**h**). (**f**) A filtered image of (**d**). One crossover repeat (37 nm) is indicated by black arrows in (**e**) and (**f**). Scale bar for (**e**) and (**f**) is shown on the right side of (**f**), 10 nm. (**g**) An averaged Fourier pattern of 15 WT actin filaments. Four prominent layer lines are indicated by arrows, showing the cross over repeat is 37 nm with a helical rise of 2.75 nm. (**h**) An averaged Fourier pattern of 18 mutant actin filaments.

**Figure 4 biomolecules-10-00736-f004:**
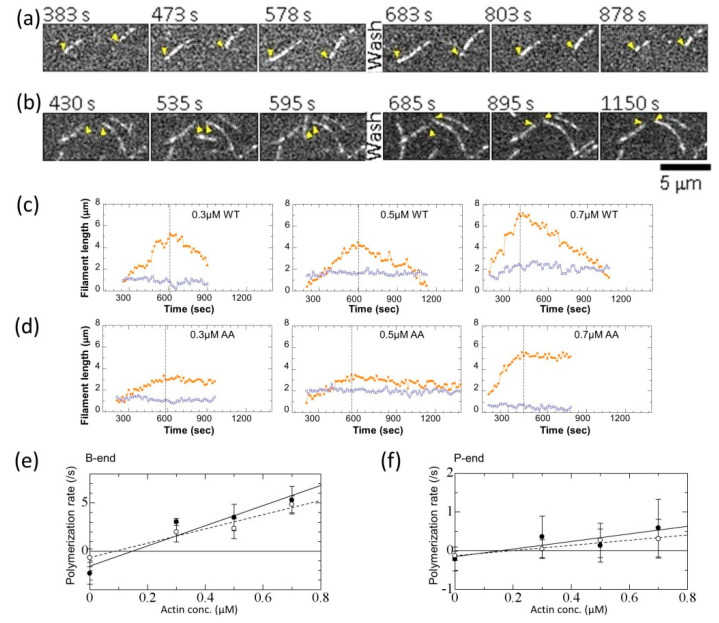
Mutation effects on actin polymerization and depolymerization. (**a**,**b**) Still images of 0.3 μM actin (20% Alexa Fluor 488-labeled skeletal muscle actin is included) polymerization under the TIRF (Total Internal Reflection Fluorescence Microscopy) assay: (**a**) wild-type and (**b**) mutant. Yellow arrowheads mark the fast growing (barbed) end of a representative filament under each condition. Time is shown in the top left corner of each image and the scale is 5 µm. Movies are published as Appendix A. The reactions were started at time 0 by mixing actin monomers with the polymerization buffer. The polymerization buffer consisted of 50 mM KCl, 1 mM MgCl_2_, 1 mM EGTA (ethylene glycol tetraacetic acid), 10 mM imidazole, 100 mM DTT, 0.2 mM ATP, 0.5% (*w*/*v*) methylcellulose, pH 7.0. Actin monomers were then removed by washing with the polymerization buffer without actin, at the time indicated by “Wash”. (**c**,**d**) Time course length change of typical recombinant actin filaments in the polymerization and depolymerization phases for actin concentrations of 0.3, 0.5 and 0.7 μM. Length changes from the barbed- (orange) and pointed-ends (purple) of each actin filament are shown: (**c**) wild-type and (**d**) mutant. Dashed lines indicate the time when actin monomers were removed by washing. After the wash, the actin filaments started to depolymerize. (**e**,**f**) Concentration-dependent average rates of polymerization and depolymerization. Rates of wild-type (filled symbols) and mutant (open symbols) at each actin concentration were estimated by fitting the length change of more than 10 actin filaments on each phase. Bars represent standard deviations (SD). Rates and critical concentrations determined by linear fitting (solid lines for WT and dotted lines for AA) are shown in Table 1. (**e**) At the barbed-end (B-end) and (**f**) at the pointed-end (P-end).

**Figure 5 biomolecules-10-00736-f005:**
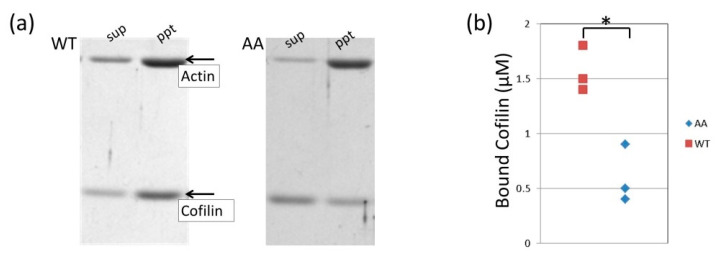
Cofilin binding. (**a**) Examples of SDS-PAGE gels for the cofilin-binding assay. Two micromolar cofilin and 2 μM actin were mixed in cofilactin buffer (15 mM PIPES (1,4-Piperazinediethanesulfonic acid)-KOH, 50 mM KCl, 2 mM MgCl_2_, 2 mM ATP, pH 6.6), incubated for 90 min at room temperature and separated by ultracentrifugation into pellet (ppt) and supernatant (sup). (**b**) The concentration of bound cofilin from three independent experiments for wild-type (WT) and the G42A/G46A mutant (AA) were determined. Statistically significant: * *p* < 0.01.

**Figure 6 biomolecules-10-00736-f006:**
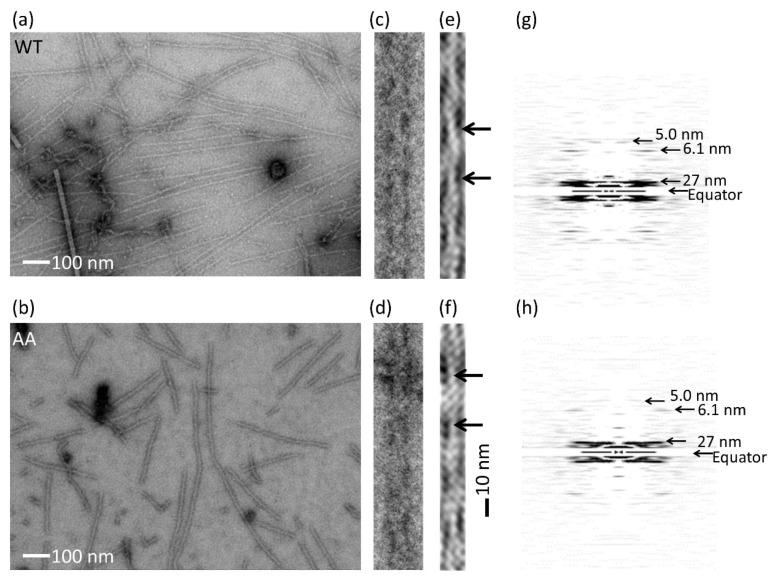
The structure of cofilactin. (**a**) An example of an electron micrograph of the wild-type (WT) cofilactin filament. (**b**) An example of an electron micrograph of the G42A/G46A mutant (AA) cofilactin filament. (**c**) An enlarged view of the WT cofilactin filament. (**d**) An enlarged view of the AA cofilactin filament. (**e**) A filtered image of (**c**). (**f**) A filtered image of (**d**). One crossover repeat (27 nm) is indicated by black arrows in (**e**) and (**f**). (**g**) An averaged Fourier pattern of 15 WT cofilactin filaments. The four prominent layer lines shown by arrows indicate that the cross over repeat is 27 nm with a helical rise of 2.75 nm. (**h**) An averaged Fourier pattern of 16 mutant cofilactin filaments.

**Table 1 biomolecules-10-00736-t001:** Elongation and dissociation rates of wild-type (WT) and mutant β-actin at barbed (B)- and pointed (P)-ends.

	WT		Mutant	
	B-end	P-end	B-end	P-end
*k*^+^ (µM^−1^ s^−1^) ^a^	11.0 ± 0.8 ^c^	1.1 ± 0.2	7.1 ± 0.6	0.7 ± 0.1
*k*^−^ (s^−1^)	1.8 ± 0.3	0.19 ± 0.08	0.6 ± 0.2	0.13 ± 0.06
Cc (µM) ^b^	0.17 ± 0.03	0.18 ± 0.09	0.09 ± 0.03	0.2 ± 0.1
N ^d^	46	56	52	50

^a^ Elongation rates (*k*^+^) were determined from fitted slopes and dissociation rates (*k*^−^) were from the *y*-intercepts (see Figure 4e,f). ^b^ Critical concentrations (Cc) were estimated from the *x*-intercepts (Figure 4e,f). ^c^ The values after the ± symbol represent standard errors. ^d^ Number of measurements.

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
