# Peer review of "D-Loop Mutation G42A/G46A Decreases Actin Dynamics"

_biomolecules, 2020, doi:10.3390/biom10050736_

Round 1

Reviewer 1 Report

The main point of this manuscript is characterising the function of the D-loop of actin molecule. Thus the authors have introduced two mutations G42A/G46A. These mutations did not cause significant effect on subunit-subunit interactions, caused small but significant difference in the critical concentration of actin, did not cause significant difference in the structure of actin filament,  caused significant reduction in binding to cofilin but the structure of cofilactin was not affected by the mutation.

The data presented in this paper sounds too preliminary in nature and require additional experiments.

1) The authors have cited Poor sample yields as the reason for not determining doing the critical concentration by light scattering analysis or the pyrene fluorescent assay. The critical concentration for the barbed ends and poined ends must be measured accurately before this paper can be considered for publication.

2) The authors used labelled rabbit skeletal muscle actin together with WT or mutant human b‐actin for determining the polymerization and depolymerization rates. The authors should label recombinant WT and mutant human b‐actin to carry out this experiment.

3) All the data is based on in-vitro assays which is can be easily affected by the buffer conditions. Thus the authors should express GFP-tagged WT or mutant human b‐actin in animal cells and visualize the actin filaments to characterize the actin filaments in the cells. 

4) The authors must determine whether there is any difference in the cofilin induced severing of actin filaments made of WT and mutant actin.

5) The authors must use cell extract to determine if there is any difference in cofilin sedimentation with WT and mutant actin.

Author Response

We colored updated descriptions in red.  

Comment

1) The authors have cited Poor sample yields as the reason for not determining doing the critical concentration by light scattering analysis or the pyrene fluorescent assay. The critical concentration for the barbed ends and poined ends must be measured accurately before this paper can be considered for publication.

Response

Slow depolymerization rates hamper accurate determination of the critical concentration for each end. The errors of the rates are similar to the following recent literature. We have performed a sedimentation assay to compare the critical concentrations more accurately (Fig. 2).

Wioland, H.; Guichard, B.; Senju, Y.; Myram, S.; Lappalainen, P.; Jegou, A.; Romet-Lemonne, G. Adf/cofilin accelerates actin dynamics by severing filaments and promoting their depolymerization at both ends. Current biology : CB 2017, 27, 1956-1967 e1957.

Fujiwara, I.; Takeda, S.; Oda, T.; Honda, H.; Narita, A.; Maeda, Y. Polymerization and depolymerization of actin with nucleotide states at filament ends. Biophysical reviews 2018, 10, 1513-1519

Comment

2) The authors used labelled rabbit skeletal muscle actin together with WT or mutant human b‐actin for determining the polymerization and depolymerization rates. The authors should label recombinant WT and mutant human b‐actin to carry out this experiment.

Response

We agree that it is better to use labeled recombinant actin for obtaining more precise measurements. However, we believe that the observations using labeled rabbit skeletal muscle actin is valid for comparison between different actin species.

Comment

3) All the data is based on in-vitro assays which is can be easily affected by the buffer conditions. Thus the authors should express GFP-tagged WT or mutant human b‐actin in animal cells and visualize the actin filaments to characterize the actin filaments in the cells. 

Response

We agree that performing experiments in cells would be important. Expressing mutants in mammalian cells is possible. However, very careful investigation is required to interpret results because it is very difficult to destroy all intrinsic actin genes. Therefore, mutant actin will copolymerize with intrinsic actin in the host cells with a much higher ratio than the current in vitro experiments, which complicates accurate analysis and interpretation of the results substantially. It would take a substantial amount of effort and time to perform these experiments and they are also outside the scope of the current work presented in the manuscript. Thus, we appreciate the comment raised by the reviewer and will consider these experiments as a future project.

Comment

4) The authors must determine whether there is any difference in the cofilin induced severing of actin filaments made of WT and mutant actin.

Response

We agree that measuring the severing effects of cofilin represents a valuable approach to further investigate the mutant. We have discussed the possibility to perform the experiments; however, universities are closed in Japan because of COVID-19. The current situation does not permit us to perform extra experiments immediately, and it remains unclear when we will be allowed to re-enter our laboratories. We believe that it is worth reporting our current results although the additional experiment would have improve the quality of our manuscript.

Comment

5) The authors must use cell extract to determine if there is any difference in cofilin sedimentation with WT and mutant actin.

Response

The cell extract contains a large variety of actin binding proteins that affect cofilin binding and activity. We believe purified cofilin is much better to use when characterizing the effects of cofilin.

Reviewer 2 Report

The paper Matsuzaki et al. presents interesting results on the effect of double mutation of D‐loop mutation on actin dynamics. The experiment appears well planned and executed. The manuscript is well organized. Some changes are recommended:

Line 40-43: The statements do not provide a comprehensive picture on the structural organization of F-actin. Because of the lack of important information regarding the F-actin architecture, one can easily consider that the information provided here is not correct. The authors should spend more effort in explaining the differences between G-actin and F-actin.

Line 73: The non-native sequence is not underlined.

Cation of Figure 1 needs to be edited. Some spaces are missing.

Abbreviations should be defined when first used in the manuscript body.

Line 253: I suggest reconsidering the title of the sub-section 3.2 in agreement with the other sub-sections

Line 384: Taking into account that is the beginning of Discussion section, it should be stated here that D-loop belongs to actin.

Section 4. Discussion: The discussion of the results against literature is rather superficial.

The impact of the findings should be stated in the Conclusion section.

Author Response

We colored updated descriptions in red.

Comments
Line 40-43: The statements do not provide a comprehensive picture on the structural organization of F-actin. Because of the lack of important information regarding the F-actin architecture, one can easily consider that the information provided here is not correct. The authors should spend more effort in explaining the differences between G-actin and F-actin.

Response

We would like to thank the referee for carefully reading our manuscript. We added Figure 1e to explain the differences between G- and F-actin.

Comments

Line 73: The non-native sequence is not underlined.

Cation of Figure 1 needs to be edited. Some spaces are missing.

Abbreviations should be defined when first   used in the manuscript body.

Line 253: I suggest reconsidering the title of the sub-section 3.2 in agreement with the other sub-sections

Line 384: Taking into account that is the beginning of Discussion section, it should be stated here that D-loop belongs to actin.

Response:

We have made those changes to the manuscript.

Comment:
Section 4. Discussion: The discussion of the results against literature is rather superficial.

Response

We have added a paragraph to discuss the results in more detail.

Comment:
The impact of the findings should be stated in the Conclusion section.

Response:

We have answered this comment.

Reviewer 3 Report

This is a very good study that addresses the role of two Gly residues in actin dynamics. The study is completely in vitro, but employs cutting edge technology and irreproachable biochemical experiments. The authors must have been disappointed that the results were not more black and white, because the only important result they get is a modest decrease in actin polymerization at the barbed end and a modest decrease of the interaction of mutant actin with cofilin.

I have nothing to complain regarding the quality of the experiments included in this study. Perhaps, the only experiment I would request is that the authors could determine if the mutant filaments interact equally well with phalloidin, which would indicate whether mutant actin filaments display a similar stability compared to wild type. Also, it would be interesting to transfect some mammalian cells with GFP-tagged mutant vs. wild type actin and check if the polymerization rate is affected in a biologically significant system, such as a protruding lamellipodium. This is something that, if the authors do not bear the required expertise, could be done in collaboration.

However, these experiments can also be considered for future papers. I understand the current global situation regarding COVID-19 limits access to the lab (this reviewer cannot get access to his own lab!), and these experiments would extend the scope of the paper, but they are not necessary to validate the conclusions of the present study.

Author Response

We colored updated descriptions in red.

Comments

This is a very good study that addresses the role of two Gly residues in actin dynamics. The study is completely in vitro, but employs cutting edge technology and irreproachable biochemical experiments. The authors must have been disappointed that the results were not more black and white, because the only important result they get is a modest decrease in actin polymerization at the barbed end and a modest decrease of the interaction of mutant actin with cofilin.

I have nothing to complain regarding the quality of the experiments included in this study. Perhaps, the only experiment I would request is that the authors could determine if the mutant filaments interact equally well with phalloidin, which would indicate whether mutant actin filaments display a similar stability compared to wild type.

Also, it would be interesting to transfect some mammalian cells with GFP-tagged mutant vs. wild type actin and check if the polymerization rate is affected in a biologically significant system, such as a protruding lamellipodium. This is something that, if the authors do not bear the required expertise, could be done in collaboration.

However, these experiments can also be considered for future papers. I understand the current global situation regarding COVID-19 limits access to the lab (this reviewer cannot get access to his own lab!), and these experiments would extend the scope of the paper, but they are not necessary to validate the conclusions of the present study.

Response

We thank the reviewer for carefully evaluating our study. A very recent paper showed that phalloidin and jasplakinolide caused different effects to the structure of the D-loop (Pospich et al., Structure (2020) 28, 437–449 e435). It would be important to further investigate the effects of phalloidin on the D-loop structure. The investigation in cells is also important. However, this will require a very careful investigation. Structural differences to the D-loop caused by drugs are limited and effects on the dynamics might not be apparent. It is also very difficult to prevent the mutant actin from copolymerizing with intrinsic actin of the host cell when observing the mutant actin in the cell, which makes interpretation and accurate analysis of the data challenging. Thus, these experiments represent a future project.

Round 2

Reviewer 1 Report

1) My original comment "The critical concentration for the barbed ends and pointed ends must be measured accurately before this paper can be considered for publication." still stands. The authors have the all the tools to carry out the in-vitro actin polymerization assay using actin-pyrene. Sedimentation assays are only indicative of polymerization.

Reviewer 2 Report

The manuscript was improved and meets the quality required for publication.

Reviewer 3 Report

I've analyzed every answer the authors have provided to me as well as the other reviewer. I'm sorry to say that the authors have taken the high road and decided not to take into account almost any of the reviewer's comments, neither mine nor the other reviewer's.

The way I understand peer review, it's always a "negotiation", in which both authors and reviewers compromise on a middle ground. Some experiments requested are not always fair, or doable. Here, the authors have not given us an inch, and provided weak rationales to justify their decision not to provide more experimentation.

Having stated that, I stand by my original evaluation. This is not a bad study, but some of the criticisms of the other reviewer are valid and some additional experimentation should be provided.